# Pre-Treatment of Fish By-Products to Optimize Feeding of *Tenebrio molitor* L. Larvae

**DOI:** 10.3390/insects13020125

**Published:** 2022-01-25

**Authors:** María-Ángeles Romero-Lorente, Dmitri Fabrikov, Juan Montes, Elvira Morote, Fernando G. Barroso, María del Carmen Vargas-García, Ágnes Timea Varga, María-José Sánchez-Muros

**Affiliations:** 1Department of Biology and Geology, University of Almería, 04120 Almería, Spain; mrl402@inlumine.ual.es (M.-Á.R.-L.); df091@ual.es (D.F.); jml154@ual.es (J.M.); fbarroso@ual.es (F.G.B.); mcvargas@ual.es (M.d.C.V.-G.); agnest.varga@gmail.com (Á.T.V.); 2Organización de Productores Pesqueros de Almería, 04007 Almería, Spain; elvira.almeria1@gmail.com

**Keywords:** fatty acid, insect meal, EPA, DHA, fish discards

## Abstract

**Simple Summary:**

The rapid population growth and the consequent increase in demand for protein-rich foods pose a challenge for the food industry. On the other hand, resources are limited and production must be sustainable. Currently, insects are accepted by the European Union as sustainable and nutritive food for humans, consequently allowing the use of seven insect species, including *Tenebrio molitor* (Linnaeus, 1758). From a nutritional point of view, *T. molitor* is rich in protein (42.2–67.7%) with a good ratio of essential amino acids. However, their lipidic fraction lacks *n*−3 long chain polyunsaturated fatty acids (LCPUFAs), while it is rich in *n*−6 polyunsaturated fatty acids. This poses a problem for human intake, since the beneficial effect of *n*−3 polyunsaturated fatty acids and a low *n*−6/*n*−3 ratio on the prevention of cardiovascular inflammation and heart disease is well known. The present study is focused on the improvement of the fatty acid profile and proximal composition of *T. molitor* larvae through fish discard-based diets subjected to different pre-treatments to increase the fish-based diet intake.

**Abstract:**

Fish discards are organic waste with high and good-quality protein levels, as well as a fatty acid profile rich in *n*−3 LCPUFAs, mainly eicosapentaenoic acid and docosahexaenoic acid. These discards can be used as food for *Tenebrio molitor* (Linnaeus, 1758) larvae, thus increasing the nutritional value of this insect. This study focused on increasing larval acceptance of fish through different pre-treatments of the diets provided, as well as increasing the accumulation of EPA and DHA in fish-fed larvae. Four different diets were prepared: control (broiler feed), DGF50: 50% dried ground fish (*Pagellus bogaraveo*, Brünnich, 1768) + 50% broiler feed, for different periods, FGF100: 100% fresh ground *P. bogaraveo* and DUF100: 100% dried whole unground *P. bogaraveo*. Growth, mortality, proximate composition, fatty acid profile and lipid nutritional indices were determined. Larvae fed with FGF100 displayed better results among treatments, doubling the initial weight, as well as increasing their protein level and decreasing fat levels. Regarding fatty acids, eicosapentaenoic acid and docosahexaenoic acid were only detected in larvae fed with a fish-based diet for a period longer than 5 days. These results show that pre-treatment of fish-based diets causes changes in the growth and compositional parameters of *T. molitor* larvae.

## 1. Introduction

The FAO estimates that the population will grow to 9100 million by 2050 [1] which would increase the food demand, especially regarding protein- and energy-rich food. It is estimated that there will not be enough resources to feed this increased population [2]. Besides this, the current food production systems must achieve sustainability, while being able to generate foods with adequate nutritive properties.

A higher demand for meat and its derivates is also expected, mostly due to the increase in the population of developing countries [3]. Nowadays, meat makes up 15% of the energy of the human diet, while 80% of cultured land is used for livestock feed (3400 million hectares as pastures and 500 million hectares as farmland) [4]. Therefore, the estimated increase in meat intake will lead to a large increase in land used for livestock feed. All these factors drive the search for new alternative protein sources [5]. In recent years, different sustainable protein sources have been studied to be used in human and animal feeding (food and feed), such as in vitro meat [6], seaweed [7], duckweed [8], rapeseed [9], microalgae, single-cell protein [10] and insects [11].

Specifically, insects show several environmental advantages [12,13]: (1) they have higher reproduction rates and food conversion efficiencies than those of poultry, swine and beef; (2) they require less water; (3) they emit lower levels of greenhouse gases.

The Commission Regulation (EU) 2017/893 (Regulation (EU) 2017/893 of 24 May 2017 as regards the provisions on processed animal protein) allowed the production of eight insect species, including yellow mealworm (*Tenebrio molitor* Linnaeus 1758), to be included in the formulation of feeds for aquaculture. Recently, the Commission Implementing Regulation (EU) 2021/882 of 1 June 2021 authorised the placing on the market of dried *Tenebrio molitor* larvae as a novel food under Regulation (EU) 2015/2283 (Regulation (EU) 2015/2283 of the European Parliament and of the Council of 25 November 2015 on novel foods) [14].

This fact and its ease of mass rearing may be the reason why, in Europe, *T. molitor* is currently one of the most bred insects [15], and it is one of the most studied in recent years, both in fish [16,17,18] and bird feeding [19,20,21]. Other uses, such as the bioconversion of industrial residues [15] and even application as alternative organic fertilizer, have also been proposed [22].

From a nutritional point of view, insects are protein-rich ingredients; specifically, the protein content of *T. molitor* varies between 42.2% [23] and 67.7% [24] with a good ratio of essential amino acids [25]. However, their lipidic fraction lacks *n*−3 long chain polyunsaturated fatty acids (*n*−3 LCPUFAs), while it is rich in *n*−6 polyunsaturated fatty acids (*n*−6 PUFAs) [26]. This poses a problem for human intake, since the beneficial effect of *n*−3 LCPUFAs and a low *n*−6/*n*−3 ratio on the prevention of cardiovascular inflammation and heart disease is well known [27,28,29]. Nevertheless, the fatty acid profile of an insect can be modified by a diet rich in *n*−3 LCPUFAs [30].

On 1 January 2019, the European Union (EU) launched a new land-based regulation for the management of fishery discards, (Art. 15, Regulation (EU) no. 1380/2013EU-CFP 2013), that forces ships to land fish discards in port. Fish discards represent approximately 10% of the total fishery catch [31] and generate protein- and *n*−3 LCPUFA-rich organic wastes that can be used as fish meal and oil, bioactive compounds [32,33,34,35,36,37] and oils rich in polyunsaturated fatty acids (PUFAs) [37]. Nevertheless, discarded fish generates waste that is difficult to store, thus the production of these materials requires the setting up of processing plants close to the port, making it difficult to comply with this new regulation. An alternative use in areas without the necessary infrastructure and low economic yield is the use of fish discards as insect feed [24,30].

In a recent study carried out by Fabrikov et al. [24], it was found that while *Hermetia illucens* (Coleoptera: Stratiomyidae) larvae accepted fish discards, *T. molitor* larvae, due to their feeding habits, did not consume them adequately. Therefore, this work is focused on the improvement of the intake of fish discards subjected to different pre-treatments by *T. molitor* larvae, with different periods of fish-based diet feeding, in order to increase the accumulation of *n*−3 LCPUFAs. Therefore, the aim of this work is to determine the effect of the inclusion of fish discards and the way the diet is administered on the accumulation of *n*−3 LCPUFAs, mainly EPA and DHA, and the proximal composition of *T. molitor* larvae to be used as human food with increased nutritional value. 

## 2. Materials and Methods

### 2.1. Larval Rearing

Larvae of *T. molitor* were obtained from MealFood S.L. (Salamanca, Spain) and bred at the University of Almería (Almería, Spain). The experiment was carried out at 26 ± 1 °C and 65 ± 5% humidity. Five diets were prepared, a control diet and four experimental diets using fish discards):a control diet © composed of commercial broiler feed (NANTA^®^, Madrid, Spain);a diet composed of 50% fish discards (*Pagellus bogaraveo*), dried and ground, + 50% poultry feed (DGF50);a diet composed of 100% dried and ground fish (DGF100);a diet composed of 100% dried whole unground *P. bogaraveo* (DUF100);and finally, a diet composed of 100% fresh ground *P. bogaraveo* (FGF100).

The fish discards, of red seabream (*P. bogaraveo*), were provided by OPP-71 (Almería, Spain). Table 1 shows the proximal composition and fatty acid profile of commercial broiler feed and *P. bogaraveo* used to prepare the diets.

The commercial broiler and dried fish diets were mixed with water at a 1:3 (*v*/*w*) ratio.

Larvae with average initial weight of 0.066 g (instar 12-13) were fed the control diet for 3 days as a measure of adaptation to the experimental environment. The larvae were then divided into separate groups in three boxes (15 × 20 cm) containing approximately 1000 larvae with a density of 33,300 larvae cm^−2^. All groups (11 treatments) were fed *ad libitum* with the experimental diets for 21 days until slaughter by freezing. Exceptionally, DGF50 larvae feed was replaced by C feed at different times. Figure 1 shows the different periods of feeding of *T. molitor* larvae with the DGF50 diet. To carry out the analysis, larvae were dried for 48 h at 60 °C in an oven to a constant weight and ground to obtain insect meal.

### 2.2. Growth and Mortality

The initial weight (Wi) in grams of the larvae was calculated by weighing 6 batches of 50 larvae, and then averaged. The average weight of individual larvae was 0.066 g. The final larval weight (Wf) was calculated by weighing 50 larvae from each batch (in triplicate) and then averaged.

Initial (Ni) and final (Nf) numbers of larvae were obtained by dividing initial and final larval biomass by Wi and Wf, respectively. Mortality rate was obtained using the following equation:% Mortality = (1 - Nf/Ni) ∗ 100(1)

### 2.3. FA Analysis

For the analysis of fatty acids, transesterification and extraction were applied simultaneously [38,39]. The steps in the process are as follows: first, 50 mg of dried sample was weighed and placed in a test tube, 0.5 mg of nonadecanoic acid (19:0) were added as an internal standard. One milliliter of a methylation mixture (methanol:acetyl chloride 20:1 *v*/*v*) and 1 mL of *n*-hexane were added and the mixture was heated at 100 °C for 30 min, thus obtaining the fatty acid methyl esters (FAMEs). Afterwards, tubes were cooled down to room temperature, 1 mL of distilled water was added and the mixture was centrifuged (2000× *g*, 5 min). Then, the organic layer was collected for subsequent analysis by gas chromatography, in the same way as described by Guil Guerrero et al. [40]. Thus, FAMEs were analyzed on a Focus GC (Thermo Electron, Cambridge, UK) fitted with a flame ionization detector (FID) and an Omegawax 250 capillary column (30 m × 0.25 mm i.d. × 0.25 μm film thickness; Supelco, Bellefonte, PA, USA). A peak area of the internal standard was used as a reference to calculate the mass of each fatty acid in the resulting chromatograms, and the results were calculated as g/100 g DM. The relative retention factor of each fatty acid was considered for quantification as reported [41].

### 2.4. Lipid Quality Indexes

The atherogenic index (AI) and thrombogenicity index (TI) was calculated by applying the following formulas [42]:AI = [12:0 + (4 × 14:0) + 16:0]/(Σ*n*−3 PUFA + Σ*n*−6 PUFA + ΣMUFA)
TI = (14:0 + 16:0 + 18:0)/[(0.5 × ΣMUFA) + (0.5 × Σ*n*−6 PUFA) + (3 × Σ*n*−3 PUFA) + (Σ*n*−3 PUFA/Σ*n*−6 PUFA)].

### 2.5. Proximate Composition

The proximal compositions of *T. molitor* and diets used in all experiments were analyzed according to the standard procedures of the Association of Official Analytical Chemists [43], with the following specific methods: crude protein (CP) content was determined by Kjeldahl * 6.25 (AOAC, 2005; #954.01) (N * 6.25); ether extract (EE) was determined by ethyl ether extraction (Soxhlet technique) (AOAC, 2005; #920.39); the moisture was gravimetrically quantified by drying at 105 ± 0.5 °C (AOAC, 2005; #934.01), while ash was gravimetrically determined after combustion at 500 °C in a muffle furnace (AOAC, 2005; #942.05) to a constant weight. All the analyses were performed in triplicate.

### 2.6. Statistical Analyses

An ANOVA normality test was performed. Subsequently, a comparison of means (Tukey’s test) was carried out. These analyses were all been carried out using the IBM SPSS Statistics 26 software (2019).

## 3. Results

### 3.1. Viability and Growth

The mortality rate and growth of *T. molitor* larvae fed with different fish-based diets are shown in Table 2.

The mortality rate of larvae fed with C, DGF50-1 and FGF100 did not differ significantly from one another, while periods of the DGF50 diet intake up to 7 days decreased the mortality rate. Longer periods of fish-based diets for larvae feeding resulted in high mortality rates, as can be seen for DGF50-14 and DGF50-21, with 48% and 42%, respectively (*p* < 0.001).

It is worth noting that two additional diets were tested, DFG100 made only with dry ground fish, and DUF100 made with unground dry fish. The larvae from these treatments had a mortality rate of about 90% after 24 h, therefore the larvae from these treatments were removed and were not analyzed further.

As can be seen, larvae fed with C, DGF50-1, DGF50-2, DGF50-3, DGF50-5 and DGF50-7 diets did not vary in their final larval weight compared to their initial weight (0.066 g). The final larval weight decreased when larvae were fed for longer periods (14 and 21 days) with the DGF50 diet, to 0.053 and 0.050 mg, respectively. The final larval weight of larvae fed with FGF100 doubled (0.13 g) compared to the initial weight, showing a significant difference compared to other diets (*p* < 0.001).

### 3.2. Proximate Composition

Table 3 shows data of proximate composition in larvae fed with fish-based diets.

Ash levels increased when larvae were fed with fish-based diets, and FGF100-fed larvae displayed significant differences (*p* = 0.001) compared to control and other fish-fed larvae, reaching the highest percentage (9%). Ether extract (EE), which represents the lipid fraction found in larvae, showed statistically significantly lower levels (*p* < 0.001) in those larvae fed fish-based diets for longer periods (DGF50-14, DGF50-21 and FGF100). Crude protein percentage increased when fish-based diets were offered for longer periods: lower levels were observed for DGF50-3-fed larvae (51.89%) and higher levels in FGF100-fed larvae (60.91%). Longer periods of fish-based diets displayed significant differences compared to control-fed larvae (*p* < 0.001).

### 3.3. Fatty Acid Profile

Table 4 shows the fatty acid profile of larvae fed with experimental diets. Saturated fatty acids (SFAs) in larvae fed with fish-based diets did not show significant differences compared to C-fed larvae, although for short periods, DGF50-2- and DGF50-3-fed larvae showed increased SFA, displaying significant differences (*p* < 0.001), mainly due to an increase in palmitic acid (16:0, PA) up to 4.5 and 4.3 g/100 g dry matter (DM) found in DGF50-2 and DGF50-3. Myristic acid (14:0, MA) showed significant differences regarding control-fed larvae and DGF50-2- and DGF50-3-fed larvae (*p* < 0.001), and the same results were observed for stearic acid (18:0, EA). Within the groups of larvae fed with fish-based diets for periods longer than 5 days, a decrease in the amount of SFAs can be observed although no significant differences are observed between those larvae and control-fed larvae.

Total monounsaturated fatty acids (MUFAs) showed a similar trend as for SFAs, and only DGF50-2 and DGF50-3 showed significant differences regarding control and fish-fed larvae, with oleic acid (18:1n9, OA) being mainly responsible regarding the increase, although within the groups of larvae fed with fish-based diets a decrease in the amount of MUFAs can be observed (*p* < 0.001), while vaccenic acid (18:1n7, VA) was only detected in FGF100-fed larvae. Polyunsaturated fatty acids (PUFAs) showed significant differences (*p* < 0.001) for DGF50-2- and DGF50-3-fed larvae compared to control-fed larvae, and the rest of the fish-fed larvae did not show significant differences compared to the control. However, LCPUFAs were found only in larvae fed with fish-based diets for a period longer than 5 days, showing statistically significant differences among larvae fed with fish-based diets for longer than 5 five days (arachidonic acid (20:4n6, ARA) (*p* < 0.001), eicosapentaenoic acid (20:5n3, EPA) (*p* < 0.001) and docosahexaenoic acid (22:6n3, DHA) (*p* < 0.001)). ARA was only detected in FGF100-fed larvae, and EPA levels increased when larvae were exposed for longer periods to fish-based diets, displaying higher levels in FGF100-fed larvae (0.51 g/100 g DM). DHA followed EPA’s trend, displaying higher levels in FGF100-fed larvae (0.24 g/100 g DM).

### 3.4. Lipid Nutritional Indexes

Figure 2 shows the *n*−6/*n*−3 index of *T. molitor* larvae fed with control and fish-based diets. DGF50-1, -2 and -3 did not show statistical differences compared to control-fed larvae. Values decreased after feeding larvae with the DFG50 diet for 5 days. The lowest values are observed in DGF50-14- and DGF50-21-fed larvae and FGF100-fed larvae (*p* < 0.001).

For indexes of lipid quality (Figure 3), the atherogenic index (AI) did not show statistical differences among treatments (*p* = 0.271). The thrombogenicity index (TI) displayed a similar trend as observed for the *n*−6/*n*−3 index. Thus, lower values were observed for DGF50-21- and FGF100-fed larvae (*p* < 0.001).

## 4. Discussion

Nowadays, due to environmental and nutritional advantages, insects are considered as a potential source of protein for humans and livestock. Nevertheless, the fatty acid profile of terrestrial insects is poor in healthy fatty acids, *n*−3 LCPUFAs, and rich in *n*−6 PUFAs and saturated fatty acids [30,44,45]. However, the nutritional values can be modified through the diet of the insects [46], specifically, the modification of the fatty acid profile of insects has been studied in order to increase *n*−3 LCPUFA levels, mostly EPA and DHA [24,44,47]. These fatty acids are found mainly in the marine environment and human beings only can ingest them through fish and other seafood. Fish discards are a source of these fatty acids that should be kept in the food chain. Previous authors have described the storage capacity of *H. illucens* of these *n*−3 LCUFAs by feeding them with fish discards [24,47,48]. In *T. molitor,* the *n*−3 LCPUFA accumulation is lower due to a poor intake of fish, probably because of an inadequate method of administration of fish discards [24]. The present work studies if the pre-treatment of fish discards affects body composition and *n*−3 LCUFA accumulation in *T. molitor* larvae.

The results showed the ability of *T. molitor* fed with fish discards to accumulate *n*−3 LCPUFAs, although the way the fish is administered influenced the mortality, development and nutritional value of larvae. Mortality data (Table 2) showed higher values for DGF50 diet-fed larvae for more than 14 days, which results in a decrease in final biomass. This great mortality could be due to a low feed intake because the larvae only ingested the broiler feed but not the dry fish [24], and this was coupled with a large difference in diet composition, nutritional imbalance and less water availability. In fact, in this experiment, we have tested two treatments (data not shown) with 100% of discarded fish, and in both diets (100% unground dry fish and 100% ground dry fish) we have found 90% mortality after 24 h of diet administration and 100% after 48 h.

The intake of fish discards for periods longer than 7 days negatively affected the final body weight, except when the diet was administered as fresh ground fish, in which case the biomass obtained was doubled compared to any of the other treatments (Table 1). In cabbage looper (*Tricoplusia ni* Hübner) and the bertha armyworm (*Mamestra configurata* Walker) fed with a high content of EPA and DHA [49], similar results were described, showing an increase in weight in larvae fed with higher levels of EPA and DHA, but with lower levels of crude fat than seen in our experiment.

Regarding body composition, a higher percentage of ash (9.01%) was found in larvae fed with fresh ground fish (FGF100), probably due to the higher mineral content in fish compared to cereals (Table 1). The comparisons between the values of ash found for larvae fed with control or DGF50 diets for short periods of time and larvae fed with DGF50 diets for longer periods support this hypothesis. Moreover, similar results were described by [47] and [24], who reported higher ash content as the percentage of fish discards was increased.

The protein content also increased with the fish ingestion (Table 3). Similar results have been obtained by Fabrikov et al. [24] with *T. molitor* larvae fed for 14 days with *Sardinella aurita* and *Pagellus bogaraveo*. Regarding fat levels, these results agree with those published previously by Fabrikov et al. [24] for larvae of *T. molitor* fed with fish discards. Thus, decreasing fat levels were observed when fish consumption was higher in the diet, especially when the insects were fed with fresh ground fish, in this case, the FGF100 diet (Table 1). This decrease in lipid content has also been reported by Fabrikov et al. [24] in *T. molitor* fed with a diet based on 100% fish. In addition, they found an increase in protein content as observed in our experiment. These changes in larval gross composition could be due to a major utilization of fat with energetic purposes, so the percentage of body fat decreased, consequently increasing the percentage of protein content. Larvae fed with ingredients rich in carbohydrates and fiber such as oat flakes and raw vegetables, considered as the natural diet of *T. molitor* [50], displayed a higher fat percentage (42.48%) [51]. The lack of carbohydrates in fish-based diets can trigger changes in larval metabolism, using the fat to obtain energy.

Regarding the fatty acid profile, the pre-treatment of fish discards as well as the duration of fish diet administration affected the fatty acid levels. Short-term administration of DGF50 increased SFAs and MUFAs but did not cause changes in *n*−3 PUFA content, while the long-term administration of DGF50 and FGF100 diets affected *n*−3 PUFA content. As has been previously described by Fabrikov et al. and Barroso et al. [24,47], this change seems to be related to the ingestion of fish discards or the control diet (broiler feed). In the case of the DGF50 diet, the larvae can distinguish between fish and broiler feed as they are mixed in the same proportions.

Previous experiments have confirmed that the PUFA content of insects is related to dietary PUFAs [47,52,53,54,55,56,57], and our results confirm this again, obtaining a higher percentage of n-3 LCPUFAs in the insects which ingested 50% dry ground fish:50 control diet for longer periods of time, 14 and 21 days, or fresh ground fish for 21 days (Table 4).

Despite the accumulation of EPA and DHA described by various authors [24,30,47], the percentage of accumulation is low regarding diet. In insects fed with oil rich in EPA and DHA, the percentage of retention is correlated with n-3 LCPUFA content of the diet, although levels were always relatively low, less than 20% in *T. molitor* and *M. configurata* [49] or 12.1% in *H. illucens* [30]. The natural diet of terrestrial insects is usually very specific and poor and, as suggested by Colombo et al. [49], there has not been strong selection pressure to highly retain *n*−3 LCPUFAs, which results in low retention efficiencies. In *H. illucens* [30,47], an acceptable accumulation of *n*−3 LCPUFAs has been observed, especially for EPA, while the retention of DHA is low. Weers et al. [58] found low rates of accumulation of DHA in the crustacean *Daphina galeata* when fed with DHA-rich feed, suggesting that it is used as energy, since it does not play any role in insect physiology. The low level of DHA obtained in our experiment, as well as in previous experiments with *H. illucens* and *T. molitor* by other authors [24,30,47], indicates a similar situation in *D. galeata*, suggesting that DHA found in insects fed with fish discards is due to recently ingested fish in the digestive tract [30,47]. However, if we consider the review about the international recommendations on oral intake of *n*−3 fatty acids of Sanz Paris et al. [59], with the intake of 100 g of *T. molitor* (DGF50-21, and especially FGF100), the daily requirement of *n*−3 FA (0.5–2 g/day) and specifically EPA and DHA (500 mg/day) is covered.

The *n*−6/*n*−3, atherogenic (AI) and thrombogenic (TI) indices are the most used indicators of the health benefits of a lipid source. A low *n*−6/*n*−3 ratio results in relatively low inflammation and potential improvements in the prevention of cardiovascular disease [27,28,29]. The recommended value of the *n*−6/*n*−3 ratio is 4 [60]. In our experiment, *n*−6/*n*−3 values decreased with ingested fish discards from 31.4 in control diet-fed larvae to 3.7 in fresh ground fish-fed larvae (Figure 2). This decrease in the n-6/n-3 index is due to an increase in *n*−3 compounds, especially EPA and DHA. AI and TI assess the probability of the incidence of pathogenic phenomena, such as atheroma and thrombus formation. The AI did not change in any of the treatments because an increase in SFA promotes an increase in MUFA. Detected values were shown to be lower than those described for tuna, 0.69 [61]; bogues, 0.92–0.59 [62]; lamb, 0.94–1.18 [63] or 0.49 observed in pectoralis muscle of broilers [64]. On the other hand, the thrombogenic index decreased with the inclusion of fish in the insect diet due to an increase in *n*−3 compounds, showing values lower than those obtained in pectoralis muscle of broilers, 1.14 [64], or lamb, 0.95 to 1.24 [63], but higher than observed in fish, 0.33–0.29 in bogue [62] and 0.27 in tuna [61]. Nevertheless, when *T. molitor* was fed with fresh ground fish, the TI decreased to levels close to those found in fish.

In summary, the result of this experiment shows that dried fish are not well accepted by *T. molitor*. However, a suitable strategy to increase larval intake could be the administration of fresh ground fish, as larvae doubled their weight, compared to control larvae, and increased *n*−3 LCPUFA content, decreasing the *n*−6/*n*−3 ratio and TI to levels close to those found in fish. Thus, *T. molitor* raised under this feed condition could be a healthy food for humans by themselves, as well as an ingredient used in meals and for other purposes (bread, pasta, cakes, etc..

## 5. Conclusions

The inclusion of fish (dry or fresh) in *T. molitor* diets (for at least 2 weeks) implies a significant decrease in the fat content of the larvae and an increase in crude protein. In addition, it maintains the SFA content and increases the percentage of PUFAs at the cost of a decrease in MUFA. Nevertheless, the FGF100 diet is the optimal diet since the larvae had the highest final weight and the highest EPA and DHA contents. In addition, they showed a high acceptance of the diet, and low mortality. Further investigations are needed to establish the optimal feed to increase EPA and DHA levels in *Tenebrio molitor*.

## Figures and Tables

**Figure 1 insects-13-00125-f001:**
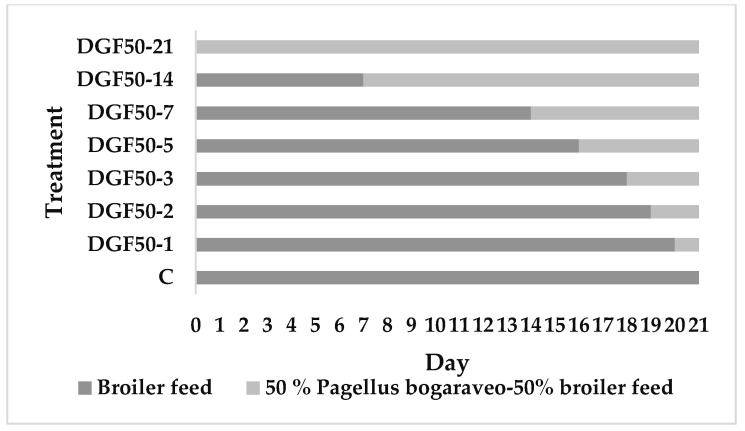
Intervals of substitution of control diet for DGF50 diet.

**Figure 2 insects-13-00125-f002:**
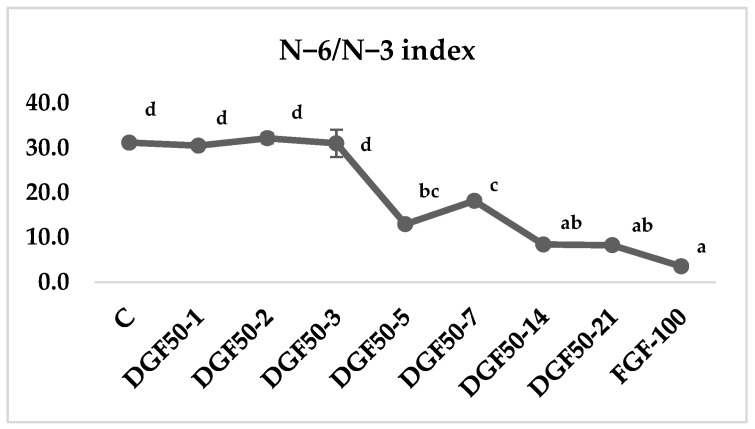
*n*−6/*n*−3 ratio of *Tenebrio molitor* larvae fed with fish-based diets. The same letter is not significantly different from one another (*p* < 0.05).

**Figure 3 insects-13-00125-f003:**
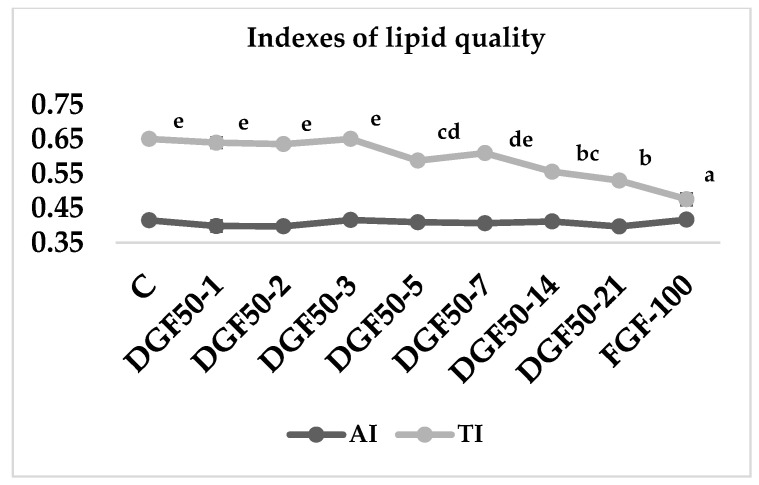
Atherogenic index (AI) and thrombogenicity index (TI) of *Tenebrio molitor* larvae fed with fish-based diets. The same letter is not significantly different from one another (*p* < 0.05).

**Table 1 insects-13-00125-t001:** Fatty acid profile (% of Fas of total Fas) and proximal composition of matter used in diets (dry matter, DM).

Fatty Acid Profile		
	Broiler Feed	*Pagellus bogaraveo*
14:00	n.d.	0.7
16:0 (PA)	0.66	3.66
16:1n7 (POA)	n.d.	1.01
17:00	n.d.	0.22
18:0 (EA)	0.11	1.52
18:1n9 (OA)	1.01	3.08
18:1n7 (VA)	n.d.	0.87
18:2n6 (LA)	2.26	0.63
18:3n3 (ALA)	0.16	n.d.
20:4n6 (ARA)	n.d.	0.28
20:5n3 (EPA)	n.d.	1.11
22:5n3 (DPA)	n.d.	0.37
22:6n3 (DHA)	n.d.	1.36
SFA	0.77	4.58
MUFA	1.01	4.96
PUFA *n*−6	2.26	0.63
PUFA *n*−3	0.16	3.12
Proximal Composition (% of DM)		
Crude protein	21.6	60.0
Ether extract	4.6	17.1
Ash	5.9	18.8

n.d.: not detected.

**Table 2 insects-13-00125-t002:** Mortality and final larval weight (g) of individual larvae of *Tenebrio molitor* fed with control and fish-based diets.

	Mortality (%)	Final Larval Weight (mg)
C	19.47 ± 0.25 ^b^	0.0621 ± 0.0002 ^b^
DGF50-1	14.55 ± 1.15 ^ab^	0.0660 ± 0.0004 ^b^
DGF50-2	10.65 ± 0.17 ^a^	0.0629 ± 0.0015 ^b^
DGF50-3	9.67 ± 0.94 ^a^	0.0648 ± 0.0021 ^b^
DGF50-5	9.67 ± 1.67 ^a^	0.0663 ± 0.0011 ^b^
DGF50-7	10.35 ± 0.89 ^a^	0.0630 ± 0.0015 ^b^
DGF50-14	48.03 ± 0.11 ^c^	0.0533 ± 0.0023 ^a^
DGF50-21	42.07 ± 1.37 ^c^	0.0507 ± 0.0010 ^a^
FGF100	13.60 ± 2.42 ^ab^	0.1314 ± 0.0008 ^c^

Means (±SE) with the same letter within a column is not significantly different from one another (*p* < 0.05).

**Table 3 insects-13-00125-t003:** Proximate composition (%) of *Tenebrio molitor* larvae fed with fish-based diets.

	Ash %	EE %	CP %
C	4.47 ± 0.08 ^a^	30.86 ± 0.65 ^d^	53.79 ± 0.27 ^bcd^
DGF50-1	4.70 ± 0.05 ^ab^	29.85 ± 0.23 ^cd^	52.26 ± 0.19 ^ab^
DGF50-2	4.36 ± 0.05 ^a^	30.35 ± 0.65 ^d^	52.91 ± 0.23 ^abc^
DGF50-3	4.52 ± 0.10 ^a^	27.55 ± 0.13 ^c^	51.91 ± 0.16 ^a^
DGF50-5	4.99 ± 0.06 ^ab^	30.38 ± 0.11 ^d^	54.55 ± 0.22 ^cd^
DGF50-7	5.02 ± 0.09 ^ab^	29.12 ± 0.18 ^cd^	54.91 ± 0.14 ^d^
DGF50-14	6.17 ± 0.40 ^b^	23.40 ± 0.51 ^b^	56.85 ± 0.87 ^e^
DGF50-21	5.10 ± 0.05 ^ab^	22.11 ± 0.10 ^b^	60.88 ± 0.21 ^f^
FGF100	9.01 ± 0.89 ^c^	19.45 ± 0.90 ^a^	60.91 ± 0.29 ^f^

EE: ether extract; CP: crude protein. Means (±SE) with the same letter within a column is not significantly different from one another (*p* < 0.05).

**Table 4 insects-13-00125-t004:** Fatty acid profiles (g FA/100 g dry weight) of *Tenebrio molitor* larvae fed with fish-based diets.

	C	DGF50-1	DGF50-2	DGF50-3	DGF50-5	DGF50-7	DGF50-14	DGF50-21	FGF100
14:0	0.5 ± 0.0 ^ab^	0.6 ± 0.0 ^bc^	0.7 ± 0.0 ^cd^	0.8 ± 0.1 ^d^	0.5 ± 0.0 ^ab^	0.6 ± 0.0 ^abc^	0.4 ± 0.0 ^a^	0.5 ± 0.0 ^ab^	0.4 ± 0.0 ^ab^
16:0	2.9 ± 0.3 ^ab^	3.7 ± 0.3 ^bc^	4.5 ± 0.2 ^c^	4.3 ± 0.2 ^c^	2.5 ± 0.0 ^a^	3.2 ± 0.1 ^ab^	2.3 ± 0.1 ^a^	2.8 ± 0.3 ^ab^	2.3 ± 0.1 ^a^
16:1n7	0.3 ± 0.0 ^a^	0.4 ± 0.0 ^abc^	0.4 ± 0.0 ^abc^	0.5 ± 0.0 ^bc^	0.3 ± 0.0 ^a^	0.4 ± 0.0 ^abc^	0.3 ± 0.0 ^ab^	0.5 ± 0.1 ^bc^	0.5 ± 0.0 ^c^
18:0	0.7 ± 0.1 ^ab^	0.9 ± 0.1 ^cb^	1.1 ± 0.0 ^c^	1.1 ± 0.1 ^c^	0.6 ± 0.0 ^a^	0.8 ± 0.0 ^ab^	0.6 ± 0.0 ^a^	0.7 ± 0.1 ^ab^	0.8 ± 0.0 ^ab^
18:1n9	7.3 ± 0.7 ^abc^	9.4 ± 0.8 ^cd^	11.6 ± 0.4 ^d^	10.9 ± 0.6 ^d^	6.4 ± 0.1 ^ab^	8.2 ± 0.3 ^bc^	5.4 ± 0.3 ^a^	7.3 ± 0.7 ^abc^	4.9 ± 0.3 ^a^
18:1n7	n.d.	n.d.	n.d.	n.d.	n.d.	n.d.	n.d.	n.d.	0.2 ± 0.0
18:2n6	4.2 ± 0.4 ^abc^	5.4 ± 0.4 ^cde^	6.5 ± 0.3 ^e^	6.2 ± 0.4 ^de^	3.7 ± 0.1 ^ab^	4.8 ± 0.2 ^bcd^	3.4 ± 0.2 ^ab^	4.6 ± 0.5 ^abcd^	3.0 ± 0.3 ^a^
18:3n3	0.1 ± 0.0 ^ab^	0.2 ± 0.0 ^bc^	0.2 ± 0.0 ^c^	0.2 ± 0.0 ^c^	0.1 ± 0.0 ^ab^	0.2 ± 0.0 ^abc^	0.1 ± 0.0 ^a^	0.2 ± 0.0 ^abc^	0.1 ± 0.0 ^ab^
20:4n6	n.d.	n.d.	n.d.	n.d.	n.d.	n.d.	n.d.	n.d.	0.2 ± 0.0
20:5n3	n.d.	n.d.	n.d.	n.d.	0.1 ± 0.0 ^ab^	n.d.	0.1 ± 0.0 ^b^	0.2 ± 0.0 ^c^	0.5 ± 0.0 ^d^
22:6n3	n.d.	n.d.	n.d.	n.d.	0.1 ± 0.0 ^a^	0.1 ± 0.0 ^ab^	0.1 ± 0.0 ^ab^	0.2 ± 0.0 ^b^	0.2 ± 0.0 ^c^
SFA	4.1 ± 0.5 ^ab^	5.2 ± 0.3 ^bc^	6.3 ± 0.2 ^c^	6.1 ± 0.3 ^c^	3.6 ± 0.1 ^a^	4.6 ± 0.2 ^ab^	3.3 ± 0.1 ^a^	4.1 ± 0.5 ^ab^	3.5 ± 0.1 ^a^
MUFA	7.6 ± 0.7 ^abc^	9.8 ± 0.8 ^cd^	12.0 ± 0.4 ^d^	11.4 ± 0.6 ^d^	6.7 ± 0.1 ^ab^	8.6 ± 0.4 ^bc^	5.8 ± 0.3 ^a^	7.3 ± 0.9 ^abc^	5.6 ± 0.4 ^a^
PUFA	4.4 ± 0.4 ^a^	5.6 ± 0.5 ^ab^	6.7 ± 0.3 ^b^	6.4 ± 0.4 ^b^	4.0 ± 0.1 ^a^	5.1 ± 0.2 ^ab^	3.8 ± 0.2 ^a^	5.1 ± 0.6 ^ab^	4.0 ± 0.4 ^a^
*n*−3	0.1 ± 0.0 ^a^	0.2 ± 0.0 ^a^	0.2 ± 0.0 ^a^	0.2 ± 0.0 ^a^	0.3 ± 0.0 ^ab^	0.3 ± 0.0 ^ab^	0.4 ± 0.0 ^bc^	0.6 ± 0.1 ^c^	0.9 ± 0.1 ^d^
*n*−6	4.4 ± 0.6 ^abc^	5.4 ± 0.4 ^cde^	6.5 ± 0.3 ^e^	6.2 ± 0.4 ^de^	3.7 ± 0.1 ^ab^	4.8 ± 0.2 ^bcd^	3.4 ± 0.2 ^ab^	4.6 ± 0.5 ^abcd^	3.1 ± 0.3 ^a^

Means (±SE) with the same letter within a row is not significantly different from one another (*p* < 0.05). SFA: saturated fatty acid; MUFA: monounsaturated fatty acid; PUFA: polyunsaturated fatty acid; n.d.: not detected.

## Data Availability

Data have been deposited in the Institutional Repository of the University of Almería.

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
