# Peer review of "Pre-Treatment of Fish By-Products to Optimize Feeding of Tenebrio molitor L. Larvae"

_insects, 2022, doi:10.3390/insects13020125_

Round 1
Reviewer 1 Report
Here you study the effect of a fish-based diet of T. monitor larvae on the growth, mortality, proximate composition, fatty acids profile and lipid nutritional indices. It is a quite interesting manuscript, however, there are major issues regarding the material and methods section (we do not know the age of the larvae that you used? were they all of the same age? How big where they?) and especially the statistical analysis section. There is no statistics presented in the entire text. You can not publish an article without presenting your statistics. Moreover, tables need to be substantially improved and finally the discussion section is too poor. Attached you will find the rest of my notes/commends concerning your manuscript.

Author Response
Thank for your comments.

Reviewer 2 Report
The paper describes the changes of fatty acid profile and proximal composition of Tenebrio molitor larvae (TML) when fed with different fish discards-based diets.
The manuscript is presented in a well-structured manner. However, misleading or wrong abbreviations of the diets for the larvae generate confusion.
Many references cited in the paper are dated and not directly related to the topic of discussion.
The article should focus on the study of TML. In several parts, it also details cases with insects very different from TM, both for eating habits and for the life cycle. It is advisable to focus the discussion only on TM or justify the reason why other insects are considered.
Recent papers used different feedstocks to improve n-6/n-3 ratio in TML (e.g. linseed as reported by Francardi et al., 2017, or seed meals in Lawal et al., 2021). No particular repercussions on the survival rate of the TML were observed. On the contrary, high mortality occurs using 2 types of fish-based diets. The paper draws unconvincing conclusions regarding the use of fish waste to enrich TML in n-3 LCPUFAs. Furthermore, these conclusions do not seem to be supported by experimental evidence (specifically, the third strategy of the conclusions section). It is strongly recommended to better argue the conclusions. Authors should also mention the further investigations needed to prove their statements.

Author Response
Thank for your comments.

Reviewer 3 Report
Dear authors, the manuscript "Pre-treatment of fish by-products to optimize feeding of Tenebrio molitor L. larvae" is interesting and worth investigation. Personally speaking, despite of well-designed, you should add more analytical data, changes on essential amino acids, please provide the standard curves (fatty acids), kjeldahl is classic nevertheless there are much more acurated alternatives, among others. Have you considered to recovery omega 3 from fish wastes, and then to add into Tenebrio formulation? Would be easier.
Regards
Round 2
Reviewer 1 Report
The manuscript has been substantially improved. Most of my comments have been addressed. However I have 2 major issues:
1) your experimental design is not appropriate. You state that before doing the experiment you didn't know the outcome or your study, that's why you included the mixed diet in your experiments. You should have done some pre-experiments.
2) your conclusion is confusing. In line 423 you state that the best diet is the 100% fish based diet. This is in contrast with what you state in line 407. In general your results are strange. According to Table 2 If you use a mixed diet then the mortality increases by increasing the duration of feeding compared to the control (100% broiler based diet). However, a 100% fish based diet is presenting significantly lower mortality, which does not make sense since according to the results T. molitor larvae do not like mixed broiler and fish based diet (higher mortality and lower final weight).
Author Response
The authors would like to thank the reviewer for his help to improve the manuscript.
Please see the attachment

Reviewer 2 Report
The revised manuscript has been improved, but there are still some inaccuracies/errors to be corrected.
I will detail corrections to be made and any comments or questions in the attached file.

Author Response
The authors would like thank to the reviewer his help to improve the manuscript.
Please see the attachement

Reviewer 3 Report
Dear authors,
The manuscript has improved. I do not agree totally "...On the other hand, the aminoacids profile seems to be a phylogenetic character (Barroso et al., 2014) depending on the specie more than diet, a disbalance of amino acids mainly affect growth which is a basic parameter of production..."
Regards
Author Response
The author would like to thank the reviewer for hid help to improve the manuscript
Please see the attachment

Round 3
Reviewer 1 Report
Based on the authors' reply to my comments, I do not have any further suggestions and thus recommend the acceptance of the manuscript in its present form.